# Religiosity and Sexual Satisfaction in Middle Age: The Moderating Role of Nonconformity

Justyna Mróz [1,*], Kinga Kaleta [1], Ryszarda Ewa Bernacka [1], Karol Kubrak [1], Małgorzata Weryszko [1] and Edyta Charzyńska [2]

[1] Department of Psychology, Jan Kochanowski University in Kielce, 25-049 Kielce, Poland;
kinga.kaleta@ujk.edu.pl (K.K.); ryszarda.bernacka@ujk.edu.pl (R.E.B.); karol.kubrak@ujk.edu.pl (K.K.);
malgorzata.weryszko@ujk.edu.pl (M.W.)

[2] Faculty of Social Sciences, University of Silesia in Katowice, 40-007 Katowice, Poland;
edyta.charzynska@us.edu.pl

[*] Correspondence: jmroz@ujk.edu.pl

**Abstract:** The period of middle adulthood presents people with self-actualization challenges related, among others, to their religious beliefs and sexual satisfaction. The aim of this study was to explore the associations between religiosity, nonconformity and sexual satisfaction, as well as to examine whether nonconformity moderates the relationship between religiosity and sexual satisfaction in middle adulthood. The following questionnaires were used: the Religious Meaning System Questionnaire, the Sexual Satisfaction Questionnaire, and the Creative Behavior Questionnaire CBQIII. A series of regression analyses using Hayes PROCESS macro was run to test the models in a sample of 667 middle adults aged between 35 and 55 ($M = 42.62$; $SD = 5.37$). The analyses revealed significant relationships between the variables. The religious meaning system showed negative correlations with both nonconformity and sexual satisfaction, whereas nonconformity was positively correlated with sexual satisfaction. An interaction effect of religiosity and nonconformity on sexual satisfaction, when controlling for gender and age, was also demonstrated. More specifically, religiosity was negatively related to sexual satisfaction in individuals with a high level of nonconformity. By contrast, for low and average levels of nonconformity, the relationship between religiosity and sexual satisfaction was insignificant. The results suggest that religiosity may interact with nonconformity when predicting the intensity of sexual satisfaction of middle adults.

**Keywords:** religiosity; religious meaning system; sexual satisfaction; nonconformity; adulthood





## 1. Introduction

Sexual satisfaction is a provocative and complicated concept. As a psychological construct, it highlights whether a person achieves fulfillment in sexual life and has a positive view of this sphere of their life (McClelland 2010; Nomejko and Dolińska-Zygmunt 2014). As such, sexual satisfaction is more than a sexual function (Dundon and Rellini 2010)—it is one of the key factors in an individual's well-being (Arenella and Steffen 2022; del Mar Sánchez-Fuentes et al. 2014; Dundon and Rellini 2010). Sexual satisfaction is predicated on many factors, including relational and psychological ones. One of them is religiosity, which gives value and meaning to different dimensions of life and is related to various indicators of well-being (Jackson and Bergeman 2011; Kim-Prieto and Miller 2018). Results of research conducted so far have revealed ambiguous associations between religiosity and sexuality (e.g., Dew et al. 2020; Higgins et al. 2010). The findings have depended on the various conceptualizations of religiosity and sexuality and different measurement tools and samples used in the studies (Bahnaru et al. 2019; Cranney 2020; Gaber and Mousa 2021; Moradi and Madani 2020), demonstrating that further exploration in this area is clearly needed.

As sexual satisfaction and general well-being may be related to age and dealing with challenges in specific life periods (Jensen et al. 2020; Kieny et al. 2020; Rosen et al. 2021), we are especially interested in the relationship between religiosity and sexual satisfaction in midlife. This period of life, although central to adulthood (Infurna et al. 2020), is related to lower satisfaction with life. Previous studies have found a U-shaped relationship between age and well-being across the lifespan, with the lowest point for life satisfaction between the mid-30s and early 50s (Blanchflower and Oswald 2008; Kieny et al. 2020; Steptoe et al. 2015). Regarding sexual satisfaction in particular, some studies suggest that age has a negative effect on it (De Ryck et al. 2012), while others indicate the opposite (Young et al. 2000). Recent investigations (Leonhardt et al. 2021), however, have revealed different trajectories of sexual satisfaction existing in marriages of middle-aged people: declining, stable, and increasing over time. Erikson (1982) has pointed out that adults should achieve a sense of intimacy and create close emotional bonds. Then, in middle adulthood, they should successfully resolve the conflict of generativity versus stagnation (Hamachek 1990; Slater 2003).The degree of fulfillment of these challenges may have an impact on one's well-being, including one's sexual satisfaction (Arenella and Steffen 2022), and be related to middle-aged people's religiosity (Myers and Williard 2003).

### 1.1. Religiosity

Religiosity may be an important component of an individual's mental life. It is generally understood as one's subjective, individual attitude towards God and the supernatural, which finds expression in one's concepts, beliefs, feelings and behaviors (Golan 2006). Religiosity has been conceptualized as religious beliefs (Pepitone and Saffiotti 1997; Skrzypińska 2002, 2014), religious commitment (Davis et al. 2010; Hayward and Krause 2013), attachment to God (Davis et al. 2008; Kent et al. 2018) an individual's religious identity (Lyons et al. 2011; Park et al. 2014), religious practices (Lyons et al. 2011; Sternthal et al. 2012), or religious coping (Park et al. 2014; Sandage and Crabtree 2012).

In our study, religiosity is conceptualized as a religious system of meaning. It is defined as an individual "system of beliefs about oneself and relationships to other people and the world, the main feature of which is an inherent relationship with the >>sphere of the sacred<< as well as orientation and meaning-creating factors" (Krok 2014, p. 26). Religious beliefs allow one to understand the events taking place in one's life and build a coherent and logical vision of the world (see Park 2007). Owing to them, one can create meaning and give meaning to various dimensions of life. Orientation and religious meaning are the two main dimensions of the religious meaning system (Krok 2011). Orientation enables one to understand the world and one's own life. On the other hand, religious meaning provides the possibility of interpreting life in terms of meaningfulness and purpose.

### 1.2. Religiosity and Sexuality

Previous research on religiosity and sexuality has demonstrated mixed findings. On the one hand, a negative relationship between these variables was shown in many studies. For instance, religiosity was related to the lower levels of sexual frequency in adults and the readiness to engage in sexual activity in young people, including adolescents (Barkan 2006; Gold et al. 2010; McFarland et al. 2011; Peri-Rotem and Skirbekk 2022; Regnerus 2005; Rostosky et al. 2004). It was also negatively related to sexual satisfaction (Higgins et al. 2010; Marcinechová and Záhorcová 2020). On the other hand, positive associations between different aspects of religiosity and satisfaction with sexual life have been observed (Arenella and Steffen 2022; Dew et al. 2020; Leonhardt et al. 2020; McFarland et al. 2011; Peri-Rotem and Skirbekk 2022).

When it comes to the religious meaning system, a study by Brudek (2017) has demonstrated that the religious meaning system was positively correlated with sexual satisfaction in older married people. His analyses revealed that people who were the most satisfied with their sex life, compared to those whose sexual satisfaction was low or extremely low, were more likely to treat religiosity as a point of reference in interpreting their internal

experiences, life events, and the rules governing social life. They viewed religious content as an aid in coping with the existential questions of individual and social life.

As shown, religiosity is linked to sexual satisfaction (Dew et al. 2020; Leonhardt et al. 2020). The relationship, however, is unclear, and the underlying mechanisms remain largely unknown (del Mar Sánchez-Fuentes et al. 2014).

The aim of the present study is to fill this gap by testing the role of nonconformity in the relationship between religiosity and satisfaction with sexual life. In middle adulthood, some variables related to independence and autonomy may be of particular importance. We expect that the level of nonconformity may be a moderator in the relationship between the religious meaning system and sexual satisfaction in midlife adults.

### 1.3. Moderating Role of Nonconformity

Nonconformity can be analyzed in four research areas: as human behavior (Bond and Smith 1996), as a behavior of social groups (Bettencourt and Sheldon 2001), as a personality structure (Jackson and Tremblay 2002), and as an aspect of creative personality (Bernacka 2020; Fürst and Lubart 2017; Grohman and Szmidt 2012; Martinsen 2011; Nęcka et al. 2006; Popek 1989, 2015; Song et al. 2012). In Popek's original theory (Popek 1989, 2015), nonconformity is defined as a relatively permanent disposition towards autonomy and independence of thought, judgment, and action, which are all based on an autonomous system of values. Nonconformity is an ability, of various intensities, to resist and oppose social pressures and the readiness to defend one's own position (Bernacka 2009, 2020; Popek 2015). It includes such characteristics as: adaptational flexibility, dominance, tolerance, self-organisation, originality, independence, openness, consistency, resilience and perseverance, activeness, courage, self-criticism, and high self-esteem (Popek 1989, 2015; see Grohman and Szmidt 2012; Nęcka et al. 2006). Nonconformity can differentiate people's self-actualization and life satisfaction (Popek 2015).

To our knowledge, nonconformity understood in this way has not been examined in the context of religious beliefs and sexual satisfaction. It should be emphasized, however, that nonconformity is especially important when a person is confronted with social pressure, norms, customs, values, and religious and political issues (see Bernacka 2020; Popek 2015). Within this context, an individual's tendency to think, evaluate and behave autonomously can influence dimensions of well-being (Kokko et al. 2013), including satisfaction with one's sexual life.

Thus the first aim of the present study was to examine the relationships between religiosity, nonconformity, and sexual satisfaction in middle adulthood (ages 35–55). We hypothesized that religious meaning system, nonconformity, and sexual satisfaction would be significantly related to each other (Hypothesis 1; H1). Since previous research has provided mixed findings, especially with regard to middle adulthood (Leonhardt et al. 2021), we cannot assume unambiguously whether there are positive or negative relationships between the variables, and so we regard our study as exploratory. Moreover, we tested whether nonconformity played a moderating role in the relationship between religiosity and sexual satisfaction. As nonconformity determines whether people tend to think and act independently or according to common norms (Bernacka 2020; Popek 2015), it may moderate the association between religiousness, understood as giving meaning and significance to various aspects of life, and the evaluation of sexual contentment. We hypothesized that this relationship would differ depending on middle-aged adults' level of nonconformity (Hypothesis 2; H2).

## 2. Materials and Methods

### 2.1. Participants

To determine the minimum sample size for the current study, G*Power version 3.1 (Faul et al. 2007) was used. The sample size required for multiple regression analyses with five independent variables to achieve a power of 0.80 to detect a low effect ($f^2 = 0.02$) at a significance level of 0.05 was 550. A total of 667 respondents in middle adulthood

(603 women and 64 men), aged between 35 and 55, participated in the current study. The mean age was 42.62 (*SD* = 5.37). The number of participants was considered suitable as it exceeded the minimum sample size calculated by G*Power. Regarding the highest level of education, 1% of the sample graduated from primary school, 2.8% from vocational education, 22.9% from secondary education, and 74% had a university degree. A total of 77.7% of the study participants lived in a city, and the remaining 22.3% lived in the country.

### 2.2. Methods

Religiosity. The Religious Meaning System Questionnaire developed by Krok (2009, 2011) measures religiosity understood in terms of meaning and two main dimensions: (1) religious orientation and (2) religious meaning. The total score of the two dimensions was used in the study. The questionnaire contains 20 statements, for example: "When reflecting on my life, I refer to religious norms." Participants responded on a 7-point Likert-type scale from 1 (*completely disagree*) to 7 (*completely agree*). In the current study, the reliability measured with Cronbach's alpha for the total score was 0.96.

Sexual satisfaction. The Sexual Satisfaction Questionnaire (Nomejko and Dolińska-Zygmunt 2014) was used to measure one's (cognitive and emotional) attitude toward one's own sexual activity. Sexual satisfaction is understood as the participant's subjective feeling that they are satisfied with their sexual life. This concept is associated with a positive attitude, i.e., positive thoughts, behaviors and emotions towards one's own sexual attractiveness, relationships and partners, as well as sex itself. The questionnaire consists of ten items, for example: "I find my sexual life fulfilling." Participants responded on a 4-point Likert-type scale from 1 (*completely disagree*) to 4 (*completely agree*). In the current study, the reliability measured with Cronbach's alpha was 0.87.

Nonconformity. Nonconformity was assessed using the Creative Behavior Questionnaire (CBQIII) (Bernacka et al. 2016). This questionnaire consists of two scales: Conformity/Nonconformity (C/N), which belongs to the sphere of personality, and Heuristic Behaviors/Algorithmic Behaviors, which belongs to the cognitive sphere. Each scale consists of 13 bipolar properties, each of which forms a dimensional continuum. The Nonconformity Scale included in the CBQIII contains 13 statements, for example: "I usually make decisions independently and am not influenced by pressure from other people." Participants responded on a 5-point scale from (score A) (*agree*) to (score E) (*disagree*). The structure of this instrument allows one to obtain scores separately for each of the scales as well as a total score measuring one's creative attitude. Nonconformity is understood as a relatively permanent disposition of varying intensity—from low to high. The higher the score on the scale, the higher the level of nonconformity (Bernacka 2020; Popek 2015). CBQIII has a test-retest reliability of 0.95. The correlation between the Polish and English versions of the C/N scale = 0.95 (Bernacka 2009; Bernacka et al. 2016). In the current study, the reliability measured with Cronbach's alpha was 0.76.

### 2.3. Procedure

We used two inclusion criteria: (1) age between 35 and 55 and (2) being in a sexual relationship. Participation in the study was voluntary; no remuneration was offered. A cross-sectional web-based survey was used to collect data. All respondents provided their informed consent online. The responses were anonymous, and the confidentiality of information was assured. Participants were informed about their right to terminate the survey at any time. The study was conducted in accordance with the Declaration of Helsinki.

### 2.4. Data Analysis

We used Pearson's correlations to assess the relationships between the main variables analyzed: religiosity, nonconformity, and sexual satisfaction. The moderation model was tested by regression analysis using IBM SPSS software (version 28, PS IMAGO PRO 6.0, Predictive Solutions) and PROCESS software (Hayes 2013), respectively. Religiosity served

as an explaining variable, sexual satisfaction as an explained variable, and nonconformity as a moderator. Gender and age were included in the model as covariates. Model 1 (moderation analysis) PROCESS was estimated with 5000 bootstrap samples and 95% bias-corrected bootstrap intervals for the moderating effect.

## 3. Results

### 3.1. Descriptive Statistics and Correlations between the Variables

In the first step of the analysis, we examined the correlations between a religious meaning system (orientation and meaningfulness), nonconformity, and sexual satisfaction (see Table 1).

**Table 1.** Descriptive statistics and correlations between religious meaning system, nonconformity and sexual satisfaction (*N* = 667).

|   |   | *M* | *SD* | 1 | 2 | 3 | 4 | 5 |
|---|---|---|---|---|---|---|---|---|
| 1 | Orientation | 30.96 | 14.51 | - | | | | |
| 2 | Meaningfulness | 39.23 | 14.44 | 0.899 ** | - | | | |
| 3 | Religious Meaning System | 70.20 | 28.22 | 0.975 ** | 0.974 ** | - | | |
| 4 | Nonconformity | 34.02 | 6.94 | −0.077 * | −0.108 ** | −0.095 * | - | |
| 5 | Sexual Satisfaction | 31.12 | 5.74 | −0.086 * | −0.074 | −0.082 * | 0.365 ** | - |

** $p < 0.01$, * $p < 0.05$. *M* = mean, *SD* = standard deviation.

The religious meaning system was weakly negatively correlated with nonconformity and sexual satisfaction. Moreover, nonconformity was moderately positively related to sexual satisfaction.

### 3.2. Moderation Analysis

The moderation analysis revealed a significant moderating effect of nonconformity on the relationship between religiosity and sexual satisfaction (β = −0.09, 95% CI [−0.163, −0.025], $p = 0.007$). Specifically, religiosity was negatively related to sexual satisfaction when nonconformity was high (*M* + 1*SD*) (β = −0.14, 95% CI [−0.239, −0.041]). By contrast, the relationship between religiosity and sexual satisfaction was insignificant for low (*M* − 1 *SD*) (β = 0.05, 95% CI [−0.050, 0.149]) and average (*M*) (β = −0.06, 95% CI [−0.130, 0.012]) values of nonconformity (see Figure 1).

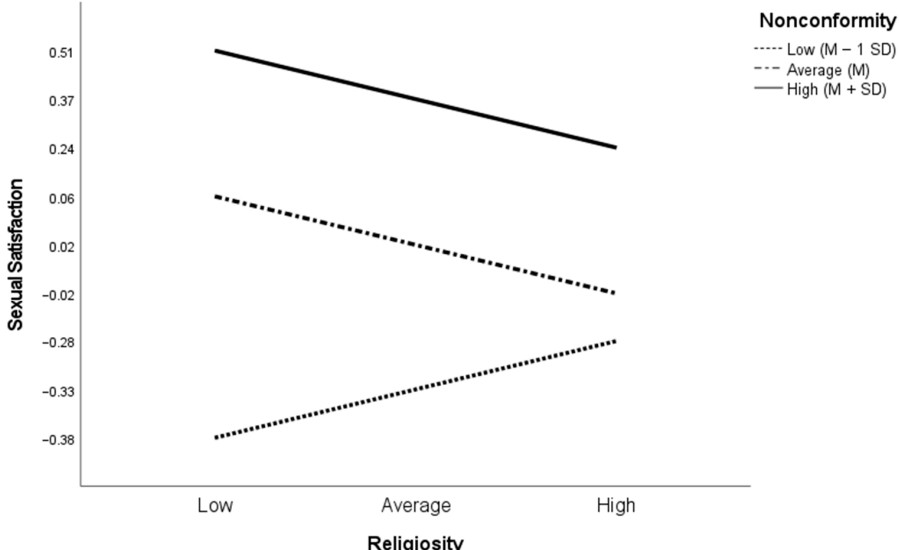

**Figure 1.** A visual representation of the moderating effect of nonconformity on the relationship between religiosity and sexual satisfaction (*N* = 667). *M* = mean, *SD* = standard deviation; explained variance ($R^2$) = 0.14.

## 4. Discussion

The present study aimed to explore the relationships between religiosity and sexual satisfaction in middle-aged people in the context of their nonconformity. The results support H1 concerning the significant relationships between religious meaning system, nonconformity, and sexual satisfaction. Religious meaning system showed weak negative correlations with both nonconformity and sexual satisfaction, whereas nonconformity positively correlated with sexual satisfaction.

Our findings are consistent with some previous studies showing a negative link between religiosity and sexuality (Marcinechová and Záhorcová 2020; Purcell 1984). The possible mechanism might be related to sex guilt (Hackathorn et al. 2016) generated by religious beliefs which disapprove some of one's sexual activities (Rostosky et al. 2004), including non-procreative sex (Beisert 2006). On the other hand, the negative correlation between the religious meaning system and sexual satisfaction was weak in our study and not consistent with Brudek's study (Brudek 2017), in which a positive association was found between these variables in older adults. This suggests that other predictors might have a stronger impact on sexual satisfaction in midlife adults, or a more complex explanation is required (see del Mar Sánchez-Fuentes et al. 2014). The results of a study by McFarland et al. (2011) suggest that the status of the relationship may play a role here: in their sample, sexual satisfaction was higher in married, religious people than in unmarried religious individuals. Moreover, religious integration in daily life showed a positive association with pleasure from sex among older married adults (McFarland et al. 2011). Rostosky et al. (2004) found that in unmarried people, religion played a prohibitive role in relation to sexual activity. Interestingly, other researchers (Peri-Rotem and Skirbekk 2022) showed that more religious married women showed greater sexual satisfaction, but in the case of religious men, those who were unmarried experienced greater sexual satisfaction.

Our results also demonstrated that the religious meaning system was weakly negatively associated with nonconformity. The possible mechanism of this association might be related to way of achieving well-being. People low in nonconformity stick to rules, avoid violating and breaking them, and tend to have a stable image of the world (Parks-Leduc et al. 2015; Purcell 1984; Roccas et al. 2002). Radoń (2010) suggests that conformity is correlated with the emotions of joy and happiness ensuing from one's own religiousness. Perhaps, this allows them to maintain a sense of security, understood as emotional and cognitive mental comfort. In turn, people high in nonconformity have a sense of psychological comfort when pursuing change and finding the unique purpose, meaning, and causes of their own actions (Popek 1989, 2015), regardless of explanations based on common religious beliefs.

The positive relationship between nonconformity and sexual satisfaction that we found might be interpreted in light of the role of the internal locus of control (Jain and Singh 2015). For nonconformists, the internal locus of control creates the basis for thinking, judging and expressing their opinions and views, including beliefs about the complexity of the social world (Chen et al. 2006) and sexual satisfaction. Nonconformity provides a sense of autonomy and independence of views regarding the meaning and purpose of human sexual behavior. As nonconformity increases autonomy and emotional authenticity, it may enhance the tendency to manifest preferred sexual behaviors, which, in turn, results in a sense of sexual effectiveness and is reflected in higher sexual satisfaction (see Jain and Singh 2015; Popek 2015).

Our second goal was to test the moderating role of nonconformity in the relationship between religiosity and sexual satisfaction. The findings we obtained revealed that religiosity was negatively related to sexual satisfaction only in individuals exhibiting higher levels of nonconformity. Such an association did not occur in the more conformist respondents (i.e., those with low or average levels of nonconformity). Being highly religious and highly nonconformist at the same time may lead a person to experience personal conflict between the rules of faith and their own needs and preferences associated with sexuality, which may negatively influence their sexual satisfaction. Such persons may experience a sense of guilt (see Hackathorn et al. 2016; Jokela et al. 2011; van Leeuwen and Mace 2016) or

moral doubts stemming from experiencing incongruence between their own sexual needs or behavior and their own moral (including religious) rules (Lewczuk et al. 2021).

From the practical point of view, when working on sexual issues with religious clients, it is important to consider their levels of nonconformity. It may create the opportunity for the therapist to identify the client's emotional conflicts caused by discrepancies between religious beliefs and a high need for autonomy and independence. This may serve as a basis for improving the client's sexual well-being.

*Limitations*

It is important to note the limitations of this study. First, the respondents were recruited randomly via the Internet using the "snowball" method. This type of research design is associated with a greater risk that the group of people participating in the study may not be representative. Second, the sample was mostly female whereas research has shown that gender differentiates both religiosity (Francis and Penny 2013) and sexual satisfaction (Freihart et al. 2021). Future studies with an equal gender ratio should be conducted to minimize the effects caused by gender imbalance. Third, we did not take into account the participants' other sociodemographic characteristics, such as marital status, having children, level of education, and different periods of life. For example, previous studies demonstrated differences in sexual satisfaction between married and unmarried people (Kislev 2020). Therefore, it is important to include more socio-demographic variables that may be relevant to the variables under study in future research. Fourth, it was a cross-sectional correlational study. Thus, the causes and effects of the phenomena discussed in it could not be determined. In future studies, researchers might use data from longitudinal studies (e.g., the daily diary method) or an experimental design. Fifth, some of the correlation coefficients of the results noted in this study were low. Therefore, research in this area should continue to verify the results presented in this article. Finally, in future research, it would also be worth including participants representing different cultures and religious beliefs, as these factors might have an important effect on the results.

**Author Contributions:** Conceptualization J.M., K.K. (Kinga Kaleta), R.E.B., K.K. (Karol Kubrak) and M.W.; methodology, J.M., K.K. (Kinga Kaleta), E.C.; formal analysis, J.M.; investigation, J.M., K.K. (Kinga Kaleta). R.E.B.; resources, J.M., K.K. (Kinga Kaleta), R.E.B., K.K. (Karol Kubrak) and M.W.; data curation, J.M., R.E.B.; writing—original draft preparation, J.M., K.K. (Kinga Kaleta), R.E.B., K.K. (Karol Kubrak), M.W., E.C.; writing—review and editing, J.M., K.K. (Kinga Kaleta), R.E.B., K.K. (Karol Kubrak) and E.C.; visualization, J.M.; project administration, J.M. All authors have read and agreed to the published version of the manuscript.

**Funding:** This research received no external funding.

**Institutional Review Board Statement:** The study was conducted in accordance with the Declaration of Helsinki. Participation in the study was anonymous and voluntary.

**Informed Consent Statement:** Informed consent was obtained from all subjects involved in the study.

**Data Availability Statement:** The data that support the findings of this study are available from the corresponding author [J.M] upon request.

**Conflicts of Interest:** The authors declare no conflict of interest.

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
