# Peer review of "Religiosity and Sexual Satisfaction in Middle Age: The Moderating Role of Nonconformity"

_religions, doi:10.3390/rel14030377_

Round 1

Reviewer 1 Report

88 - Insert "has"

131-135 - I would suggest the inclusion of two supporting references, one at the end of each of the two sentences.

172-174 - The wording of this sentence requires attention. I would suggest that the word "highest" is included: "Regarding the highest level ..."

180 and 184  - Global? I am unsure what global means in these sentences.

180 - "The global score ..." this could be rewritten to be more clear? Perhaps "The total score of the two dimensions ..."

184 - "the global score ..." this could be rewritten to be more clear? Perhaps "the total score of the two dimensions ..."

184 - Insert "with"

200 - Should the name of the scale be written using leading capital letters? I.e., "The Norconformist Scale"?

217 - Correct typological errors by moving "the Declaration of Helsinki up a line removing the gap. 

235-236 Table 1 - The bold lines in Table 1 are in places incomplete.

235-236 Table 1 - Error "-.082*" should be "-0.082*" 

321 - Query "on the sexual issue" could be reworded as "on sexual issues"

338 - Should be "e.g.,"

341 - Replace "Finally" with "Fifth"

342 - Replace "In future research ..." with "Finally, in future research ..."

References - Query the (in)consistent translation of text from Polish to English.

Author Response

First, we would like to thank the Reviewers for the time devoted to reading our article. We would also like to thank you for your valuable comments and useful suggestions that helped us improve the quality of our paper. The response to the Reviewers’ comments and suggestions can be found below. All changes in the manuscript are marked in red.

Reviewer #1

Suggestions for Authors

88 - Insert “has”

Following the Reviewer’s suggestion, we have inserted “has.”

131-135 - I would suggest the inclusion of two supporting references, one at the end of each of the two sentences.

We have included the references.

172-174 - The wording of this sentence requires attention. I would suggest that the word "highest" is included: "Regarding the highest level ..."

We have included the word “highest.”

180 and 184  - Global? I am unsure what global means in these sentences.

We have meant “the total score of the two dimensions.” In the revised manuscript, we have rewritten this phrase.

180 and 184- "The global score ..." this could be rewritten to be more clear? Perhaps "The total score of the two dimensions ..."

We have changed the phrase as suggested.

184 - Insert “with”

We have inserted “with.”

200 - Should the name of the scale be written using leading capital letters? I.e., "The Norconformist Scale"?

We have corrected it as suggested.

217 - Correct typological errors by moving “the Declaration of Helsinki up a line removing the gap.

We have changed it as suggested.

 235-236 Table 1 - The bold lines in Table 1 are in places incomplete.

We have corrected it.

235-236 Table 1 - Error "-.082*" should be "-0.082*"

We have corrected it.

321 - Query “on the sexual issue” could be reworded as “on sexual issues”

We have corrected it.

338 - Should be “e.g.,”

We have corrected it.

341 - Replace “Finally” with “Fifth”

We have changed the word as suggested by the Reviewer.

342 - Replace “In future research ...” with “Finally, in future research ...”

We have added the suggested word.

References - Query the (in)consistent translation of text from Polish to English

We have corrected the References and completed the Polish references with an English translation

Reviewer 2 Report

Line 33 "predicated on..."

Line 34 - some would argue that religiosity is not an integral component

Lines 53-54 "midlife marriages" meaning those that occur in midlife or people in marriages as they pass through mid-life?

Line 58 "separation-individuation" what does that mean? Separation is the child separating from parents. Individuation is Jung's process of self-realization and spiritual development. It appears you are conflating the two. Lines 54-60 are speculative, generalizations, and step outside of the scope of the research, adding unnecessary variables. 

through 66 "developmental tasks" is outside of what you need in this paper to do your tasks. If, at the end you postulate that such tasks may also have an impact on sexuality and religiosity, it would make sense but be unnecessary. Developmental stages are theoretical and not essential to your task, and an enormous subject to tackle. 

68 Religiosity is not "an integral component of an individual's mental life", unless, for a particular individual, it is. Not for atheists, agnostics, or people who believe in avoiding religious participation like the plague. Even if it is a religious individual, it is not "integral to" or a part of, but is accepted into, or influenced by. There is a big difference. This is a broad brush assumption and generalization.

94-113 Clearly marriage context is a different ballgame than single life. 

119-124  Whoa! Personality? Is this a primary variable? What is "personality-level"? Can someone non-conform at another level? Conformity has to do with the individual's relationship to society, and is not a personality decision. It is a behavioral choice with multiple bases. It is not a level of personality. No idea what you are talking about here. 

131 Non-conformity is a behavioral choice based on personal discretion as the individual relates to a social group, as a social group relates to other social groups, as an individual or social group relates to the whole of humanity. It can range from a quirk to an obsession, can be momentary or life-long. 

140 Mystical religious experience is NOT religiosity. Religion is a historical tradition, with doctrine, liturgy, rituals, and social cohesion, among other things. Mystical experience has nothing to do with religion, but can come through a religious experience. Mystical experience is a thoroughly individual, internal experience that has to do with the internal spiritual consciousness and experience of the individual's spiritual development. It has absolutely nothing to do with whether the individual conforms to external entities or not. 

212 "being in a romantic relationship"? Really? One can be in such a relationship with absolutely no sexual or religious participation. Now we jump to the romantic experience? What will this tell us about the relationship between sexuality and religiosity? No impact of marriage? No impact of being single, just a "relationship"? Is this an affair? If an extramarital affair there is far more going on than sexuality and religiosity. What exactly are you talking about, and what exactly are you studying? 

253 Religiosity, sexuality, and nonconformity, are not personality constituents. They are behavioral choices. You are not studying personality. You are studying behavioral choices and whether or not those choices are related to or influence one another. 

260 Are you studying the relationship between religiosity and sexuality? Or are you studying nonconformity as a mediator? Religiosity by definition is conformity - conforming one's self to a religious tradition. 

274 "religious meaning system" can be what the person feels about his/her religiosity, or it can be the meaning taught by the religion itself. 

305 What does "stress " have to do with any of this? That is another consideration altogether. 

311 A nonconformist may want nothing whatsoever to do with sex at all because they want total autonomy. A hermit living in the woods is a radical nonconformist, but is highly unlikely to participate in any sexual relationship. 

341 Yes it is a big problem that respondents were "mostly female", which skews your findings to show what mostly women think, which, we all know, can be completely different from what men think. 

Limitations you discuss are all big enough to devalue your results. The primary limitation of this study is that it is all over the place with definitions, the focus shifts from sentence to sentence, and it is full of unnecessary and ancillary considerations that, although are nice to discuss, make this unclear and too thick to get through successfully. 

You have the makings of a good study, but you must narrow it into what exactly you are studying, which sounds like the issue of nonconformity as a possible mediator between sexual satisfaction and religiosity. But then you must discuss how nonconformity relates to both sexual participation and religious participation. Things like personality and mystical experience should be cut out altogether because they are ancillary. Cut out the dead wood, clarify your primary topic, and stick to it. Save discussion and speculation for the end. Clarify and be consistent with your terms, such as religion or religiosity or religious participation. They are not the same, and if you float from one to another you have nothing when you arrive wherever it is you are going. 

This could also be a very important study, but it has to be cleaned up, clear, methodology appropriate and effective, and clear presentation of results. State your topic, describe it in consistent, clear terms, and never waiver from it, unless you identify additional subjects for discussion at the end. 

Author Response

First, we would like to thank the Reviewers for the time devoted to reading our article. We would also like to thank you for your valuable comments and useful suggestions that helped us improve the quality of our paper. The response to the Reviewers’ comments and suggestions can be found below. All changes in the manuscript are marked in red.

Reviewer #2

Line 33 “predicated on...”

As suggested, we have replaced “predicted by” with “predicated on.”

Line 34 - some would argue that religiosity is not an integral component

We agree with your suggestion, so we have removed this strong claim.  

Lines 53-54 “midlife marriages” meaning those that occur in midlife or people in marriages as they pass through mid-life?

By “midlife marriages,” we meant marriages of middle-aged people.  We have reformulated this wording as follows: “...different trajectories of sexual satisfaction existing in marriages of middle-aged people.”

Line 58 “separation-individuation” what does that mean? Separation is the child separating from parents. Individuation is Jung’s process of self-realization and spiritual development. It appears you are conflating the two. Lines 54-60 are speculative, generalizations, and step outside of the scope of the research, adding unnecessary variables.

We have removed this redundant paragraph.

through 66 “developmental tasks” is outside of what you need in this paper to do your tasks. If, at the end you postulate that such tasks may also have an impact on sexuality and religiosity, it would make sense but be unnecessary. Developmental stages are theoretical and not essential to your task, and an enormous subject to tackle.

We have shortened this paragraph, omitting the unnecessary term “developmental tasks.”

68 Religiosity is not “an integral component of an individual’s mental life”, unless, for a particular individual, it is. Not for atheists, agnostics, or people who believe in avoiding religious participation like the plague. Even if it is a religious individual, it is not “integral to” or a part of, but is accepted into, or influenced by. There is a big difference. This is a broad brush assumption and generalization.

We have changed the wording as suggested. The sentence has been changed to read as follows „Religiosity may be an important component of an individual’s mental life”

94-113 Clearly marriage context is a different ballgame than single life.

We have included a paragraph about marital status in the Discussion section.

119-124  Whoa! Personality? Is this a primary variable? What is “personality-level”? Can someone non-conform at another level? Conformity has to do with the individual’s relationship to society, and is not a personality decision. It is a behavioral choice with multiple bases. It is not a level of personality. No idea what you are talking about here.

131 Non-conformity is a behavioral choice based on personal discretion as the individual relates to a social group, as a social group relates to other social groups, as an individual or social group relates to the whole of humanity. It can range from a quirk to an obsession, can be momentary or life-long.

We adopted Popek’s (1989, 2015)  approach to nonconformity, conceptualizing it as a disposition, not a particular behavior. However, to better meet the Reviewer's expectations, in the revised manuscript, we have removed the confusing wording and provided a clearer definition of nonconformity.

140 Mystical religious experience is NOT religiosity. Religion is a historical tradition, with doctrine, liturgy, rituals, and social cohesion, among other things. Mystical experience has nothing to do with religion, but can come through a religious experience. Mystical experience is a thoroughly individual, internal experience that has to do with the internal spiritual consciousness and experience of the individual’s spiritual development. It has absolutely nothing to do with whether the individual conforms to external entities or not.

We have removed this paragraph so as not to confuse the concepts.

212 “being in a romantic relationship”? Really? One can be in such a relationship with absolutely no sexual or religious participation. Now we jump to the romantic experience? What will this tell us about the relationship between sexuality and religiosity? No impact of marriage? No impact of being single, just a “relationship”? Is this an affair? If an extramarital affair there is far more going on than sexuality and religiosity. What exactly are you talking about, and what exactly are you studying?

We agree with the Reviewer that the phrase “being in a romantic relationship” was misleading. In the revised manuscript, we have replaced the phrase “romantic relationship” with “sexual relationship.”

253 Religiosity, sexuality, and nonconformity, are not personality constituents. They are behavioral choices. You are not studying personality. You are studying behavioral choices and whether or not those choices are related to or influence one another.

We agree with this comment. We have corrected misleading wordings.

260 Are you studying the relationship between religiosity and sexuality? Or are you studying nonconformity as a mediator? Religiosity by definition is conformity - conforming one’s self to a religious tradition.

We studied relationships between religiosity, nonconformity, and sexual satisfaction. Moreover, we tested whether nonconformity plays a moderating role in the relationship between religiosity and sexual satisfaction.

274 “religious meaning system” can be what the person feels about his/her religiosity, or it can be the meaning taught by the religion itself.

We agree with the Reviewer’s comment. In our research, we followed the Krok’s (2014) approach, which treats “religious meaning system” as an individual “system of beliefs about oneself and relationships to other people and the world, the main feature of which is an inherent relationship with the >>sphere of the sacred<<, as well as orientation and meaning-creating factors” (Krok 2014, p. 26).

305 What does “stress” have to do with any of this? That is another consideration altogether.

The paragraph about the relationship between nonconformity and stress has been removed.

311 A nonconformist may want nothing whatsoever to do with sex at all because they want total autonomy. A hermit living in the woods is a radical nonconformist, but is highly unlikely to participate in any sexual relationship.

We have removed this misleading phrase.

341 Yes it is a big problem that respondents were “mostly female”, which skews your findings to show what mostly women think, which, we all know, can be completely different from what men think.

We agree with the Reviewer that a mostly female sample is a limitation of our study. However, we have partially addressed this problem by adjusting the model for gender.  Moreover, we have highlighted in the Limitation section that the next study should include more male participants.

Limitations you discuss are all big enough to devalue your results. The primary limitation of this study is that it is all over the place with definitions, the focus shifts from sentence to sentence, and it is full of unnecessary and ancillary considerations that, although are nice to discuss, make this unclear and too thick to get through successfully.

You have the makings of a good study, but you must narrow it into what exactly you are studying, which sounds like the issue of nonconformity as a possible mediator between sexual satisfaction and religiosity. But then you must discuss how nonconformity relates to both sexual participation and religious participation. Things like personality and mystical experience should be cut out altogether because they are ancillary. Cut out the dead wood, clarify your primary topic, and stick to it. Save discussion and speculation for the end. Clarify and be consistent with your terms, such as religion or religiosity or religious participation. They are not the same, and if you float from one to another you have nothing when you arrive wherever it is you are going.

This could also be a very important study, but it has to be cleaned up, clear, methodology appropriate and effective, and clear presentation of results. State your topic, describe it in consistent, clear terms, and never waiver from it, unless you identify additional subjects for discussion at the end.

We appreciate the Reviewer’s thoughtful comments and suggestions. We agree that the first version of our manuscript included too many redundant terms, aspects, considerations, etc. In the revised manuscript, we have narrowed the study, clarified the topic, and used the terms in a more consistent way.

References

Krok, Dariusz. 2014. Religijny System ZnaczeÅ„ i Poczucie Sensu Å»ycia Jako Predyktory Eudajmonistycznego Dobrostanu Psychicznego u Osób Chorych Na Nowotwór [The religious meaning system and the meaning in life as predictors of eudaimonistic well-being among people with cancer]. Studia Psychologica 14 (2): 25–40.

Popek, StanisÅ‚aw. 1989. Kwestionariusz Twórczego Zachowania KANH [KANH III Questionnaire of Creative Behaviour]Lublin: UMCS.

Popek, StanisÅ‚aw. 2015. W krÄ™gu aktywnoÅ›ci twórczej [In the Circle of Creative Activity]. Lublin: UMCS.

Round 2

Reviewer 2 Report

Greatly improved from first submission, much cleaner, clearer, more succinct, and readily understandable. Congratulations on a fine study and good contribution!